# Decoding N400m Evoked Component: A Tutorial on Multivariate Pattern Analysis for OP-MEG Data

**DOI:** 10.3390/bioengineering11060609

**Published:** 2024-06-13

**Authors:** Huanqi Wu, Ruonan Wang, Yuyu Ma, Xiaoyu Liang, Changzeng Liu, Dexin Yu, Nan An, Xiaolin Ning

**Affiliations:** 1Key Laboratory of Ultra-Weak Magnetic Field Measurement Technology, Ministry of Education, School of Instrumentation and Optoelectronic Engineering, Beihang University, 37 Xueyuan Rd., Haidian Dist., Beijing 100083, China; whqmeg@buaa.edu.cn (H.W.); by1917076@buaa.edu.cn (R.W.); ma_yuyu@buaa.edu.cn (Y.M.); liangxiaoyu17@buaa.edu.cn (X.L.); by2143115@buaa.edu.cn (C.L.); 2Hangzhou Institute of National Extremely-Weak Magnetic Field Infrastructure, 465 Binan Rd., Binjiang Dist., Hangzhou 310000, China; 3Shandong Key Laboratory for Magnetic Field-Free Medicine and Functional Imaging, Institute of Magnetic Field-Free Medicine and Functional Imaging, Shandong University, 27 South Shanda Rd., Licheng Dist., Jinan 250100, China; ydx0330@sina.com; 4Hefei National Laboratory, Gaoxin Dist., Hefei 230093, China

**Keywords:** OP-MEG, modality fusion, MVPA, decoding pipeline, MI

## Abstract

Multivariate pattern analysis (MVPA) has played an extensive role in interpreting brain activity, which has been applied in studies with modalities such as functional Magnetic Resonance Imaging (fMRI), Magnetoencephalography (MEG) and Electroencephalography (EEG). The advent of wearable MEG systems based on optically pumped magnetometers (OPMs), i.e., OP-MEG, has broadened the application of bio-magnetism in the realm of neuroscience. Nonetheless, it also raises challenges in temporal decoding analysis due to the unique attributes of OP-MEG itself. The efficacy of decoding performance utilizing multimodal fusion, such as MEG-EEG, also remains to be elucidated. In this regard, we investigated the impact of several factors, such as processing methods, models and modalities, on the decoding outcomes of OP-MEG. Our findings indicate that the number of averaged trials, dimensionality reduction (DR) methods, and the number of cross-validation folds significantly affect the decoding performance of OP-MEG data. Additionally, decoding results vary across modalities and fusion strategy. In contrast, decoder type, resampling frequency, and sliding window length exert marginal effects. Furthermore, we introduced mutual information (MI) to investigate how information loss due to OP-MEG data processing affect decoding accuracy. Our study offers insights for linear decoding research using OP-MEG and expand its application in the fields of cognitive neuroscience.

## 1. Introduction

Over the past 20 years, the use of MVPA techniques in analyzing neural time-series data has exerted significant impact on the field of neuroscience [1,2]. The introduction of decoding methods provides the possibility to unfold information processing in human brain with increased sensitivity compared with univariate methods [3]. Decoding is the process of predicting outcomes using a trained model based on data, and it is the primary application of Multivariate Pattern Analysis (MVPA) [4,5]. Numerous decoding studies in cognitive neuroscience are based on fMRI due to its high spatial resolution and data quality [6,7].

MEG is a powerful electrophysiological tool with high temporal resolution, which means that neural activities can be resolved over the timescale of milliseconds [8], faster than fMRI. However, to our knowledge, MVPA studies with MEG are not as numerous as those with fMRI. Decoding techniques for time series neuroimaging data have witnessed application in the development of brain–computer interfaces (BCIs) and cognitive neuroscience through SQUID-MEG [9,10]. Despite this, traditional SQUID-MEG has some limitations, such as increased stand-off distance between sensors and the brain, and its fixed helmet constrains the subjects. Recently, quantum sensing has found applications in various fields, including inertial measurement, the detection of extremely-weak magnetic fields, and notably in MEG detection [11,12].

Many studies have pointed out the potential use of OP-MEG in unconstrained conditions [13,14], and the effectiveness of OP-MEG has been verified in detecting event-related components under unmoving [15] and moving conditions [16], which facilitates MVPA on naturalistic cognition research. Dash et al. [17] compared the performance of magnetometers and gradiometers in decoding imagined and spoken speech from the MEG signals of healthy participants. The findings suggested that while gradiometers remained preferable for MEG-based decoding analysis, OP-MEG could be considered for the development of next-generation speech-BCIs. Teplansky et al. [18] investigated OP-MEG’s feasibility in detecting auditory tones using linear discriminant analysis (LDA) and a support vector machine (SVM) with four different kernels, and proved its robustness in decoding neural activities. More recently, the single-trial decoding performance of SQUID-MEG and OP-MEG was compared, the study revealing that the OP-MEG system provides high-fidelity data about the brain with room for improvement in high band-width non-invasive BCI [19]. Bezsudnova and Jensen [20] discussed the optimization of magnetometer arrays and pre-processing pipelines for MVPA in the context of OP-MEG data. The study underscored the significance of selecting appropriate noise-reduction techniques and sensor configurations to enhance the performance of MVPA in MEG data processing. Another study revealed the decoding difference of EEG and OP-MEG through constructing temporal generalization matrix (TGM, a MVPA method) at the sensor- and source-level [21]. From the studies above, conducting decoding research in an unconstrained state is an important research direction for the future deconstruction of brain activity patterns.

The purpose of this work is to create a decoding tutorial guide for analyzing OP-MEG time series data within the field of cognitive neuroscience. Our research primarily focuses on the factors affecting the time-resolved decoding performance of evoked components obtained through OP-MEG recordings. While there have been studies on brain–computer interface (BCI) decoding using SQUID-MEG [22,23], there is currently a lack of tutorial guide specifically for interpreting temporal brain activity using OP-MEG, and our work aims to fill that gap. However, the primary aim of BCI involves translating neural activity directly into control commands for external devices, whereas for cognitive neuroscience, neural decoding is frequently employed to comprehend how the brain processes information, makes decisions and forms higher cognitive functions such as memory and emotions. Time-resolved decoding of OP-MEG evoked components can help us in understanding the temporal dynamics of stimulus input from the aspects of neuropsychology (here, we are investigating the temporal dynamics of the brain’s response to auditory stimuli for semantic congruity). Thus, BCI may attach great importance on overall optimal prediction accuracy and generalizability, which is not aligned with decoding in neuroscience [24]. Since decoding in BCI and cognitive neuroscience has different aims [25,26] and OP-MEG is superior over SQUID-MEG in channel information capacity and SNR [15,27] with a smaller number of channels, we provide a tutorial introduction using an OP-MEG N400 evoked response data with statistical inference across subjects on single time points or small time windows. In this way, our work enables subsequent research utilizing time-resolved decoding to observe the dynamic changes in brain activity during the processes of comprehending and generating language. This, in turn, provides an analytical strategy for studies investigating the temporal dynamics of language processing, yielding results with higher consistency and resolution.

Raw multi-channel neural data usually contain redundant information. Classifiers tend to be overfitted to noise and require more computation time if provided with excessive features [28]. Consequently, selecting an appropriate dimensionality reduction (DR) method prior to multivariate decoding becomes especially crucial. The Principal Component Analysis (PCA) method is a data-driven approach that transforms data into linearly uncorrelated components, arranging them based on the variance they explain [29]. This ability to separate artifacts in different components favors classification tasks for simple linear models. Prior to multivariate analysis, some studies prefer to select MEG channels with significant stimulus-related information using univariate analyses [10]. Furthermore, studies have demonstrated that compared to not performing DR, using PCA and Analysis of Variance (ANOVA) to select channels of interest significantly enhances classification accuracy [24]. By selecting features that have the most significant differences, univariate methods can effectively reduce the dimensionality of the dataset while retaining important discriminatory information.

OP-MEG is often sampled at wide bandwidth in low SNR, and one simple way to enhance it is to ‘collapse data’. This procedure is also advantageous when conducting multivariate pattern analysis (MVPA) with high temporal resolution. There are two primary methods to realize this through sliding window [30] or resampling the data to lower frequencies. The difference between these two methods lies in the fact that the sliding window collapse data by averaging and overlapping. On the other hand, resampling does not average data, but it reduces computation time since it uses fewer time points.

The choice of decoder is an essential step in the MVPA analysis of OP-MEG. Different machine learning models make different assumptions about the data. Unlike machine learning studies, MVPA in neuroscience tends to favor the selection of decoders that are simple and easy to interpret [24]. Linear decoders determine a certain label by setting a threshold based on the weighted summation of the measured responses [31]. However, linear decoders require that the distribution pattern is somewhat linearly separable. Linear decoding fails when information is encoded in a more complex form [32]. Teplansky et al. [18] decoded OP-MEG data with LDA and non-linear models, and it turned out that similar decoding performance was evident across decoders; while, in most studies concerning decoding auditory evoked components, linear models, e.g., SVM [33] and Logistic Regression [34], are usually more favored. Iivanainen et al. [19] compared the decoding performance of responses to three different sound frequencies over time under a linear discriminant model, using dual-axis OPM and SQUID, as well as a multi-class model combining EEGNet with xDAWN spatial filtering, Riemannian geometry, and Logistic Regression. Both OP-MEG and SQUID-MEG showed classification effects for binary and multi-class conditions that were significantly better than chance level. Furthermore, the results revealed that within the same local acquisition area, the classification performance of OP-MEG was superior to that of SQUID-MEG, but both were slightly inferior to full-head SQUID-MEG.

In addition to testing approaches for the decoding process, we also need to select a classic psychological evoked component as the subject of our study. The N400 component, associated with semantic processing, has a counterpart in MEG identified as N400m, which reaches its maximum around 400 ms following the onset of the stimulus [35]. Studies have indicated that N400 can reflect the retrieval of lexical information from long-term memory to working memory [36]. The component can be effectively decoded and can serve as a reference for the study of decoding evoked components under other types of paradigms.

Above all, this study will provide a wide overview of existed methods for the subsequent OP-MEG decoding studies to refer. Thus, we avoid mathematical formulations and the rationale of decoding models, and instead we focus on the principle behind different decoding approaches. Accordingly, the objectives of this article are to (a) illustrate how decoding methods impact OP-MEG data, (b) show how various analysis parameters affect the outcomes of OP-MEG decoding and (c) discuss what factors led to these influences.

The remainder of this article is structured as follows: First, we explain to the paradigms, experimental system, data preprocessing and the parameter settings of various processing approaches related to MVPA decoding performance in the Materials and Methods section. The processing approaches mainly unfolded in three major aspects: data transformation and dimension reduction, SNR improvement and decoding and test methods. Additionally, we introduce the mutual information (MI) methods and statistical methods used in our work. Second, in the Results section, we independently present the effects of different data-processing approaches and analysis options on the final decoding curve. We analyzed the impact of data dimension reduction using different methods on the amount of shared information and subsequently on decoding accuracy. Third, we discuss our research findings in the Discussion section.

## 2. Materials and Methods

In this section, we describe the methods for the paradigm, OP-MEG data acquisition, preprocessing and analysis. For preprocessing, we compare the effects of resampling, DR, sliding windows and averaging trials on the decoding results. For analysis, we compare classifiers, cross-validation and modalities on the decoding performance. And inference statistical methods were applied on the results to achieve across-subject analysis and find the underlying factors that affect the results.

### 2.1. Participants

Six right-handed subjects with normal hearing and no history of neurological diseases participated in this study (four male subjects and two female subjects, 28 ± 1.67 years old). Ethical approval for the experiment was issued by ethical Committee of Beihang University (Nr.BM20200175) and all the volunteers gave their written consent.

### 2.2. Paradigm

This experiment adopted a classic Chinese N400 paradigm reported by Ye et al. [37]. Participants were presented with Chinese sentences and were asked to give judgement to the congruity of sentence-final verbs. As shown in Figure 1a, this experiment consists of 196 trials. All sentences were declarative. In each trial, the participants first heard the part of the sentence excluding the verb at the end of the sentence. After playing this part, they waited for one second before they heard the final verb to the sentence main body. Afterwards, the subjects waited for a hint sound to give the left and right key operation. The right key and the left key represented right and wrong, respectively. During the waiting process, the subjects needed to look at the red cross in front of them. Each sentence under the two conditions was played in a pseudo-random sequence. Each sentence main body was played once under two conditions. There were at least 30 trials between two repetitions of the same key verb (one for congruent use and another for incongruent use). There were no more than four consecutive playbacks of the same stimulus type.

### 2.3. Experiment System

The digital paradigm was implemented on (http://psychtoolbox.org/, accessed on 12 June 2024) based on MATLAB 2016b (MathWorks, Natick, MA, USA) Psychtoolbox-3. The sound is transmitted from the Etymotic ER-3B headset through the silicone tube into the MSR. The residue magnetism level in the MSR is 5.5 nT. As shown in Figure 1c, the subjects were in the central position of the shielding cabin, and they were told that they needed to keep their whole body stable during the MEG acquisition. The sound stimulation was synchronized with the MEG data acquisition system.

The OP-MEG system is consisted of forty 2nd-generation QuSpin zero-field magnetometers (Quspin Inc., Louisville, CO, USA) respectively with a typical sensitivity < 15 fT/$\sqrt{\mathrm{Hz}}$.

The EEG system uses 64-channel BrainCap (Brain Products DmbH, Munich, Germany) under the 10–20 system. EEG signals were sampled at 1000 Hz, and online 0.01–250 Hz filtering was performed. The EEG electrode was re-referenced offline to the average electrode of both sides of the mastoid. The impedance was maintained below 30 kΩ.

### 2.4. Data Preprocessing

For OP-MEG data, bad channels were removed according to power spectral density (PSD) levels. Then, the raw MEG data were subjected to mean field correction (MFC) to suppress external interference [38]. Next, we applied a notch filter with a bandwidth of 2 Hz to suppress the narrow-band interference of 50, 63, 70, and 77 Hz-frequencies at which we observed peaks in the power spectral density among all subjects. Then, we used a low-pass filter with a cutoff frequency of 30 Hz to enhance OP-MEG data. Epochs containing large muscle artifacts were regarded as bad by computing z-score value, and the threshold was data dependent and was set after checking the data distribution by plotting the converted raw epochs data. Then, independent component analysis (ICA) was performed on the data to eliminate irrelevant interference. Baseline correction was performed on the epochs. For EEG data, we utilized the EOG channels (‘FP1’, ‘FP2’) as references to eliminate the most significant ICA components. Then, we performed offline high-pass digital filtering on the continuous EEG data with a cutoff frequency at 1 Hz. Then, we used the EOG channel (‘FP1’, ‘FP2’) as references to remove the most relevant ICA components. Then, we applied offline digital filtering to the continuous EEG data, using a cut-off frequency of 30 Hz. Epochs were set from 200 ms pre-stimulus to 1000 ms post-stimulus. Epochs were identified as bad according to the z-score criterion. Finally, ICA denoising and baseline correction were performed on the epochs under two conditions.

### 2.5. Analysis Summary

The impact of alternating one parameter while keeping others constant on the decoding results were explored. We should note that the parameters are not entirely independent, which means that there may be interactions affecting the analysis outcomes. Thus, following decoding outcomes should be regard as explanatory. In Figure 2, we summarize the decoding choices we investigated in this work. The effects of different choices on the decoding results were plotted as function of classifier accuracy over time. All the comparative works in this paper are based on this default pipeline.

MVPA Preprocessing (see MVPA Preprocessing section):-Subsampling 200 Hz.-Single-trial analysis.-PCA retaining 99% of the variance.-time-wise analysis, no sliding window.Decoding (see Decoding section):-Linear Discriminant Analysis (LDA) classifier.-10-fold cross-validation.-OP-MEG single modality.

### 2.6. MVPA Preprocessing

MEG data are often contaminated by various types of noise, such as environmental noise, baseline activity and physiological movements like eye blinks, which are several orders higher than target signal components [39]. To enhance signal quality, a set of standardized procedures is employed to improve SNR. Moreover, MEG data have a high dimensionality, leading to the usual practice of reducing the analysis to a fewer number of dimensions. Specifically, in MEG decoding, this reduction often involves reducing the number of spatial features (channels) and observations (trials) and applying temporal smoothing. Various methods for these MVPA preprocessing steps are commonly adopted and introduced in this section.

#### 2.6.1. Data Transformation and DR

Adjacent MEG channels typically contain a significant amount of redundant information, which can lead to overfitting effects for certain classifiers if too many features are provided. Thus, it is a common practice to reduce the data dimensionality through feature selection before decoding, which can be achieved in various ways. One method is to select the most informative channels using model-driven univariate analysis [40]. In contrast, one can also use unsupervised data-driven approaches, which transform data into uncorrelated components. The components are ranked by the amount of variance they explained [29]. In our work, as for the univariate test, we applied channel-wise permutation *t*-test on the within-subject epochs to pick out the channels that showed true effects (*p* < 0.05) between two conditions during the interested time period, i.e., 200–600 ms. And the epochs data of selected channels underwent a further decoding process.

For the data-driven DR operation, we applied PCA method on the channel dimension, for which we retained components that could explain 99% of the variance after the principal component decomposition of the standardized data. And the PCA transformation was computed on the training data and applied on the training and test data separately. The reason for choosing components with 99% variance explained is because Grootswagers et al. [24] retained components that accounted for 99% of the variance, which reduced the dimensionality of SQUID-MEG data with 160 channel dimensions to an average of 48.16.

#### 2.6.2. Improving SNR

OP-MEG data are often sampled at high frequencies, e.g., 1000 Hz, and we commonly improve the SNR through collapsing data. As mentioned above, there are two main approaches to realize this goal: sliding window or down-sample data. To this, we resampled the OP-MEG data to 200, 100 and 50 Hz and compared the decoding accuracy with no resampling. Similarly, we performed a sliding window average on the OP-MEG data with window lengths of 5, 10, 20 ms, and a step size of 0.6 times the window length. We then compared the decoding results with those obtained from the data without the sliding window averaging.

Similarly, we can increase the SNR by averaging trials under the same conditions before decoding. Studies have shown that the averaging process can generally enhance decoding performance and make signatures more pronounced [10]. Thus, we compared the impact on decoding accuracy of averaging OP-MEG data over every 2, 4, and 6 trials.

### 2.7. Decoding and Test

We performed a decoding test on the processed OP-MEG data. To decode the mental representation process for Chinese final-verb congruity, a pattern classifier was trained to distinguish congruent conditions from incongruent conditions in a time-resolved manner. A classifier’s ability to generalize to new data was evaluated through cross-validation, and the classification accuracy was regarded as the decoding accuracy. MEG signal pattern contained class-specific information once the classifier accuracy based on cross-validation was significantly above chance probability, i.e., 0.5. In this way, we can assert that the task-relevant component (N400 in our study) is significantly decoded. Time-resolved decoding on OP-MEG was repeated on each time point to generate the decoding curve.

#### 2.7.1. Decoder

From the aspect of machine learning, decoders are varied across several types, such as linear classifiers, non-linear classifiers and probabilistic model, etc. In this section, we compared the decoding performance over time with two linear decoders (LDA and Ridge Regression) and one nonlinear decoder (SVM with a ‘Sigmoid’ kernel function). The LDA classifier works by finding a linear combination of features that separates two or more classes of objects or events as much as possible. Ridge regression solves a regression equation where the loss function is the linear least squares function and regularization is given by the l2-norm. To compare linear model with non-linear model, we introduced SVM model with ’Sigmoid’ kernel, which could map the feature space to a higher-dimensional space, allowing the construction of a nonlinear decision hyperplane to it.

Before the analysis, we first balanced the number of epochs across two conditions, since balancing the number of trials under two conditions can enable the classifier to better learn the characteristics of the data under both conditions. OP-MEG data were then standardized, and the dimensions were reduced by PCA retaining 99% variance components, as mentioned above. The LDA classifier was trained using the parameters: singular-value decomposition solver, no priors, no shrinkage and tolerance (1 × 10^−4^). SVM classifier was trained using parameters such as: ‘Sigmoid’ kernel, tolerance (1 × 10^−4^), shrinkage, no class weight. Ridge regression model was trained as: ‘Stochastic Average Gradient descent’ solver, and the parameter alpha, which controls the regularization strength, is set to 1.

#### 2.7.2. Cross-Validation

We conducted a comparative analysis using cross-validation, employing stratified K-fold (k = 2, 5, 10) cross-validation with a training-to-testing data ratio of K-1:1. The stratified K-fold guaranteed that each fold contains the same percentage of samples of each class as the whole dataset. We obtained the average classification accuracy by averaging the decoding curves yielded from the k-fold cross-validation.

#### 2.7.3. Modality

To investigate the decoding performance of EEG, OP-MEG and the fusion of both modalities, we compared the accuracy curves of the three modalities data under default settings. OP-MEG and EEG data were decoded separately according to the default options. To accurately integrate the two modalities of data and neutralize the impacts arising from disparate scale unit, the fusion process necessitates an initial equalization of the trial counts for the subjects across both conditions, following the normalization of two modality data to ensure scaling to a uniform variance. The combined decoding to the fused data requires first balancing the number of trials for the respective subjects under the two conditions, and then standardizing the different modality data to scale to unit variance. The classification approach was time-resolved, with pattern vectors created from the MEG and EEG sensors separately for every millisecond.

### 2.8. Mutual Information

To investigate the impact of information loss and decoding accuracy resulting from DR and data collapsing, we introduce MI as a metric to calculate the quantity of retained information after each data operation. We measure information loss by calculating the MI difference between the OP-MEG data without DR (benchmark data) and with DR at three dimensions (operated data). Our specific method involves

Fitting the distributions of the benchmark data and the operated data with Gaussian mixture models. Here, we determined the optimal number of distributions using the Akaike Information Criterion (AIC).Sampling on the two fitted data distributions using Monte Carlo methods. We calculated the probability of the sampled points in their respective distributions to estimate the mutual information using the ‘k-nearest neighbor’ method.Averaging the computed mutual information indices to obtain the final result.Repeating the above process for each participant’s data.

#### Statistical Analysis

The non-parametric Wilcoxon signed-rank test [41] was applied in our study for examining whether within-subject classifier performance is significantly above chance (one-sided right tail, α = 0.05), since it makes minimal assumptions on the distribution of the data. To pick out the interested channels through univariate inference methods for DR use on participant-level, we performed channel-wise paired samples *t*-test (two-sided, α = 0.05) on OP-MEG data and corrected the *p*-value using the Benjamini–Hochberg method (q = 0.05) to address multiple comparison problem [42]. Correlation for the MI and accuracy difference under DR approaches at three data dimensions was computed using ’Pearson’ correlation analysis. Significance level for the correlation was based on the correlation coefficient and the number of data points under the null hypothesis that no correlation existed. We also conducted pairwise independent sample *t*-tests on information loss, i.e., MI, yielded by DR reduction in three different dimensions, in order to examine whether there was a significant difference in the loss caused by the reduction across various dimensions.

## 3. Results

Figure 3 illustrates the impact of DR operation on decoding accuracy using OP-MEG data. For such datasets and classifiers, the performance when no DR operation on OP-MEG data within the significant decoding time period exceeds that when PCA and univariate analysis methods are utilized on the data. At the 260 ms time point, the maximum decoding accuracy for the three choices occurred, which were 0.69, 0.63 and 0.58, respectively. As results for the significant decoding interval indicate, the longest significant duration is observed when no DR techniques are employed, characterized by continuous decoding from 180–210 ms and 230–640 ms, as well as intermittent decoding after 700 ms. After DR operation using the PCA method, the significant time intervals demonstrate effective noise control, manifested as continuous decoding from 180–210 ms and 230–570 ms. By univariate DR operation, the significant time periods are further shortened (240–600 ms), and intermittent decoding occur within this interval.

As we can see in Figure 4, the benefit for sliding window and subsampling on decoding OP-MEG data is marginal. The decoding performance did not show any significant differences among the choices under the two operations, and there were no marked differences in the significant decoding time periods either. However, averaging trials can result in changes in decoding performance.

As illustrated in Figure 5, we can observe that using a different number of averaged trials has an impact on the peak decoding time, decoding accuracy and the significant time intervals. The peak decoding values for no averaging, and averaging every two, four, six trials are, respectively, 0.63, 0.67, 0.66 and 0.64. The corresponding peak times are: 250, 250, 260 and 420 ms. The different numbers of averaging trials also result in varying durations of significant decoding intervals. The longest significant time intervals are observed when averaging every two trials, but the scattered decoding intervals after 700 ms demonstrate weaker control over noise. When averaging every four trials, despite better noise control after 700 ms, there is a reduced ability to decode within the range of the N400 component. When averaging over every six trials, the time period for significant decoding is most scattered, and the decoding ability is the worst of all. Here, we can infer that averaging trials affects the OP-MEG decoding performance most across all data collapsing approaches.

From Figure 6, we can see that classifiers and test choices have significant effects on decoding ability. As for the effect of classifier types on decoding, linear classifiers are generally more favored, as the LDA and ridge regression generally have better decoding ability across conditions than the non-linear model, i.e., SVM with ‘Sigmoid’ kernel. The maximum classification accuracy of LDA and ridge regression is 0.63 at the same time (250 ms post-stimulus), while for SVM, it is 0.59 at 180 ms post-stimulus. Two different types of decoders also yield distinct decoding results during different time frames. The nonlinear decoder exhibits no significant decoding during the 430–520 ms interval, where the linear decoder has shown notable decoding; instead, it demonstrates concentrated decoding results around 600 ms. At the same time, different cross-validation methods can also affect the decoding ability of linear models on OP-MEG data. Although both 5-fold and 10-fold CV exhibit peaks in classification accuracy at 250 ms, the decoding ability with 10-fold is stronger (0.63 vs. 0.61), and 2-fold CV reaches its peak later at 300 ms with 0.6. Also, a more conservative result is yielded with 2-fold CV mostly in the range 300–520 ms. Compared with 10-fold CV, 5-fold CV produces a similar significant period during 230–570 ms, but it is prolonged at 580–720 ms.

As shown in Figure 7a, different modalities of data also exhibit distinct decoding characteristics. When compared to EEG, the OP-MEG data display a higher decoding peak value (0.63 vs. 0.61). Moreover, the two modalities differ in their decoding of neural representation patterns across various stages, as evidenced by EEG’s ability to segment the process into three phases (210–370 ms, 450–570 ms, and 630–700 ms), while the decoding interval for OP-MEG is more concentrated within 230–570 ms, whereas in Figure 7b, OP-MEG exhibits a higher classification accuracy than EEG in the 270–440 ms, 620 ms, and 720–770 ms intervals, and reaches a peak value of 0.16 at 320 ms. Interestingly, within the range of 130–800 ms, the time intervals when multimodal fusion data decoding outperform OP-MEG decoding are complementary to the intervals where OP-MEG is superior to EEG.

We further analyzed the relationship between the information loss from DR across different dimensions and the decoding accuracy. When comparing the information loss brought by DR across the three dimensions in terms of MI values (Figure 8), we find that the information loss resulting from DR along the channel dimension is significantly greater than that of the other two dimensions. As can be seen from Figure 9a, the loss of mutual information due to model-driven spatial DR is greater than that using data-driven method, and there is also greater variance among the subjects. However, the change in MI produced by the two types of driving models is generally a positive correlation with a difference in decoding accuracy. As seen in Figure 9b, the greatest information loss occurs when averaging every six trials (with the Mean MI around 1.0), leading to significant fluctuations in decoding accuracy differences across subjects. However, the overall trend remains significantly correlated. When averaging every two or four trials, the Mean MI approaches 1.5, at which point a negative correlation is observed between the independent and dependent variables. As seen in Figure 9c,d, both operations on the third dimension exhibit smaller inter-subject variability in terms of the difference in decoding accuracy and MI, showing a negative correlation between the two variables. And sliding window produced reduced variability in decoding accuracy and MI, compared with resampling. Overall, the independent variable is positively correlated with the dependent variable until about 1.5, after which they are negatively correlated.

## 4. Discussion

Facts that we need to clarify are that the purpose of time-resolved decoding is to understand brain processing through statistical inference on available information. Therefore, the differences in decoding accuracy across studies are not the focus of our work [43]. Instead, decoding results should be easy for interpretation and robust to noise in the data [44]. Previous M/EEG MVPA studies on mismatch-related semantic neural substrates have shown that this process is cascading and is made up of early (N400) and late (P600) mismatch-sensitive processes, which are partially overlapped in time [45]. Time-resolved MVPA can be used to investigate the separability of cascading components [46]. OP-MEG and EEG have played complementary roles in MVPA studies on semantic congruency matching, with OP-MEG demonstrating superior performance in capturing stable coding from the phonemic to the semantic stage and subsequent dynamic coding, while EEG excels in reflecting stable coding during phonemic and semantic processing [21].

The model-driven DR operation, despite choosing the most informative channels under univariate analysis, resulted in the poorest decoding performance. Our analysis of the correlation between the amount of information and peak decoding performance across various DR choices revealed that the information capacity is positively correlated with decoding performance. This explains why the model-driven approach performs the worst of all DR operations. In contrast, data-driven methods are capable of separating such as noise and artifacts out into their respective components, which contains information unrelated to the class labels, allowing the classifier to suppress them. As indicated by Wu et al. [21], on-diagonal significant decoding only covers 200–750 ms post-stimulus. Thus, we have reasons to believe that the significant decoding occurring around and after 800 ms is largely due to introducing uncertainty factors caused by noise. In this way, we recommend further time-resolved MVPA studies using MVPA should consider using data-driven DR operation to enhance interpretability and robustness to noise.

For the OP-MEG data, even though we observe that the effects of subsampling and sliding windows on improving decoding accuracy are marginal, it cannot be denied that the computational burden has decreased. The reason why the sliding window leads to smaller fluctuations in decoding accuracy and information loss across subjects may be due to the increased signal-to-noise ratio it brings, as well as the improvement in information loss due to the overlapping sections when the step size is smaller than the window length. Therefore, regarding the enhancement of the data SNR, the optimal decoding setup should consider the type of dataset and the temporal resolution [24]. Furthermore, we observe that the impact on the temporal DR is far less significant than that of the spatial DR concerning the decoding results of OP-MEG. But, we believe this might be related to the layout and the number of channels of the OP-MEG. Due to the large channel capacity of OP-MEG, it usually does not require the same number of sensors as SQUID-MEG, and the sensors will be placed in areas of interest to enhance sensitivity to neural signals. The loss in channel capacity resulting from DR in the channel dimension is significantly greater than that in the temporal dimension. This can explain why using subsampling and sliding windows produce different effects compared to DR operations. What is more, as Iivanainen et al. [19] indicated, the decoding accuracy obtained from OP-MEG data is higher than that from SQUID-MEG in the same scalp coverage area. Subsequent research should focus on the impact of different layouts (number of channels and/or layout configuration) on the decoding performance with DR operation on the channel dimensions, in order to investigate whether and how the spatial sampling method affects the decoding performance. In comparison, the impact of averaging trials on decoding performance is pronounced, reflected in both the onset time and the peak values. Judging from classification accuracy, averaging every two or four trials seems to be the most suitable. However, in terms of interpretability, decoding results without trial averaging are more consistent with reports in the literature. However, averaging more trials (above 4) does not seem beneficial for decoding, as this will reduce the trial number in each condition (i.e., averaging over every 6 trials will produce far less data to be decoded than over 2 and 4 trials), which generally increases subject-to-subject variability in decoding accuracy.

From the perspective of modeling, analyzing the impact of classifiers on OP-MEG decoding, linear models are more favored because they yield more interpretable results, and there are reports indicating that linear models are more consistent with the brain’s pattern of encoding external stimuli [28]. The simpler the decoders, the more sensitive to class-specific information they will be.The functioning of linear decoders is analogous to that of single neuron. [32]. Overall, LDA and ridge regression perform similarly, since LDA without shrinkage is non-sparse, whereas ridge regression is also non-sparse with an l_2_ penalty. As indicated by Varoquaux et al. [47], averaging time-resolved decoding results gives a large gain in stability for non-sparse models.

The choices of the fold number for CV also has some impact on the onset and offset timing and the accuracy of decoding. A good CV strategy should minimize discrepancy. In single-trial decoding studies, choosing more folds means that there are more decoding results available to average out the within-subject variance that arises due to a smaller number of subjects [47].

From the perspective of modality, we believe that EEG is superior to OP-MEG in differentiating evoked components within the 200–800 ms range through decoding. However, OP-MEG tends to be more sensitive than EEG to brain patterns during significant time periods of decoding. Meanwhile, when combining OP-MEG and EEG for decoding analysis, the enhancement of OP-MEG decoding capability within its sensitive time periods appears to be marginal or may even be adversely affected. Nevertheless, the integrated decoding process is capable of compensating for time intervals where OP-MEG is less sensitive but EEG is more responsive. The above findings are consistent with what the literature [21,48] has reported: that MEG and EEG are sensitive to partially common and partially unique neural representations, with different temporal dynamics.

Based on the correlation and statistical analysis results, it is observed that the greatest loss of information occurs during the DR of OP-MEG data in the channel dimension, followed by the reduction in the trial dimension, i.e., observation dimension, and lastly in the temporal dimension. According to the research conducted by Marhl et al. [49], the OP-MEG system possesses a higher channel capacity and spatial sampling ability compared to the SQUID-MEG system, which is demonstrated by the fact that it can achieve the same effectiveness with only one-fourth the number of sensors. The correlation between the amount of information and decoding performance is characterized within a certain range as initially positive and subsequently negative. This means that when the loss of information is severe (average MI < 1.5), the smaller the loss and therefore the closer the decoding accuracy is to the accuracy without any DR operation. When the loss of information is minimal (average MI < 1.5), the maximum decoding value exceeds the maximum value without any DR operation, and in this case, the greater the loss of information, the closer it is to the accuracy without any DR operation. However, there are some limitations that need further investigation. First, the degree of interpretability needs to be measured using quantifiable indicators. For example, Kia et al. [50] put forward a metric that characterized the interpretablity by calculating the cosine value between the linear brain decoding model (model parameters) and the ideal solution. In this study, we compared our work with literature reports as well as made comparisons across choices to find out consistency. Second, the interactive effects across different approaches on decoding performance are not investigated in this work, and preprocessing methods do influence the choices of classifier. Future works can involve the multifaceted comparison concerning the interactions of parameters. Third, the sample size is acceptable for single-trial decoding but limits the robustness of correlation studies. Future research will require a larger number of subjects to make the correlations more significant.

## 5. Conclusions

In this article, we performed a comparison concerning different choices under various approaches, such as preprocessing, decoders, CV and data modality, on the decoding performance of evoked components based on Chinese congruity in OP-MEG recordings. We find that DR operations in the spatial domain and the averaging of trial numbers on OP-MEG data have a significant impact on decoding results, while in the temporal domain, the DR operations have a marginal impact on results. From the perspective of information capacity, spatial DR on OP-MEG causes the severe loss of information, leading to significant changes in decoding outcomes, which indirectly indicates the advance of OP-MEG in spatial sampling. Regarding the choice of decoders, we observe that linear models outperformed nonlinear ones in decoding OP-MEG data, producing more interpretable results. The method of CV affects the onset and offset of significant decoding as well as the accuracy. Lastly, we find that data modality impacts the separability and sensitivity of evoked components, demonstrated by less separability between two adjacent components in OP-MEG but higher decoding accuracy during significant time periods compared to EEG and fused OP-MEG and EEG modalities. In all, our work provides a tutorial guide for subsequent OP-MEG decoding studies in the selection of approaches and explaining how the selection will affect the decoding results. And, we also illustrated the relationship between the information loss caused by data collapsing at each dimension and the decoding accuracy. Moreover, we reaffirm the complementary advantages of EEG and OP-MEG in decoding applications.

## Figures and Tables

**Figure 1 bioengineering-11-00609-f001:**
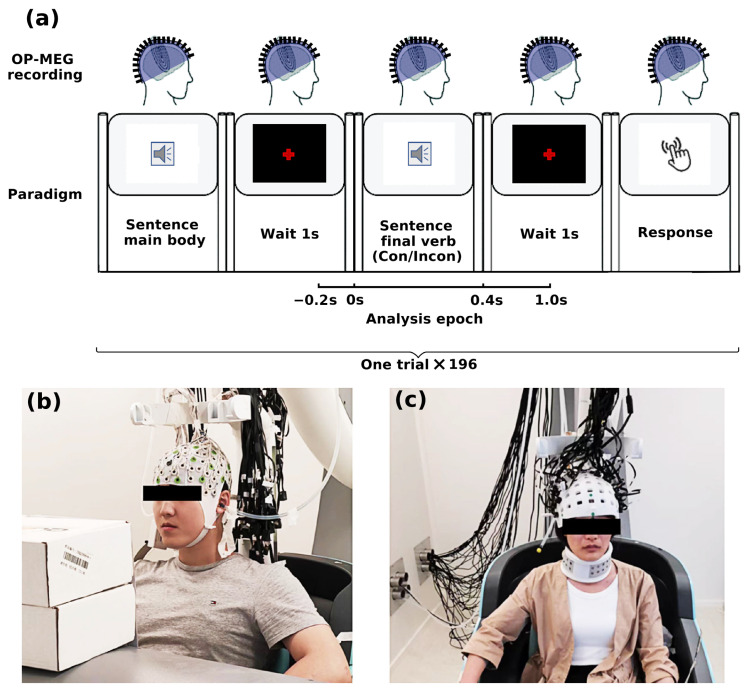
(**a**) Flowchart of the paradigm. Subjects taking the test were first presented with a sentence main body followed by the final verb after 1 s for comprehension. After another 1 s for thinking, subjects gave their response to the congruity judgement. The whole trial lasted 1.2 s in total, i.e., 0.2 s pre-stimulus and 1 s post-stimulus. The whole experiment consisted of 196 trials in all. (**b**,**c**) Subjects were sitting in the middle of MSR to take EEG and OP-MEG recording.

**Figure 2 bioengineering-11-00609-f002:**
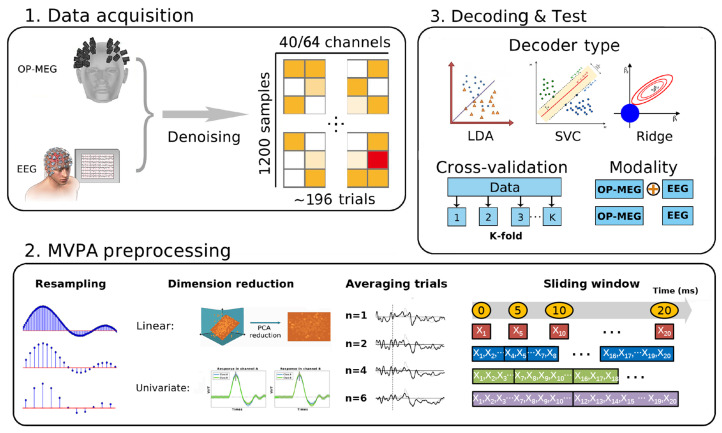
An overview for data acquisition and decoding options. In preprocessing part, the methods of improving data SNR and reducing dimension are compared. In decoding and testing parts, the decoder types, cross-validation folds and different modalities were used to systematically evaluate decoding performance.

**Figure 3 bioengineering-11-00609-f003:**
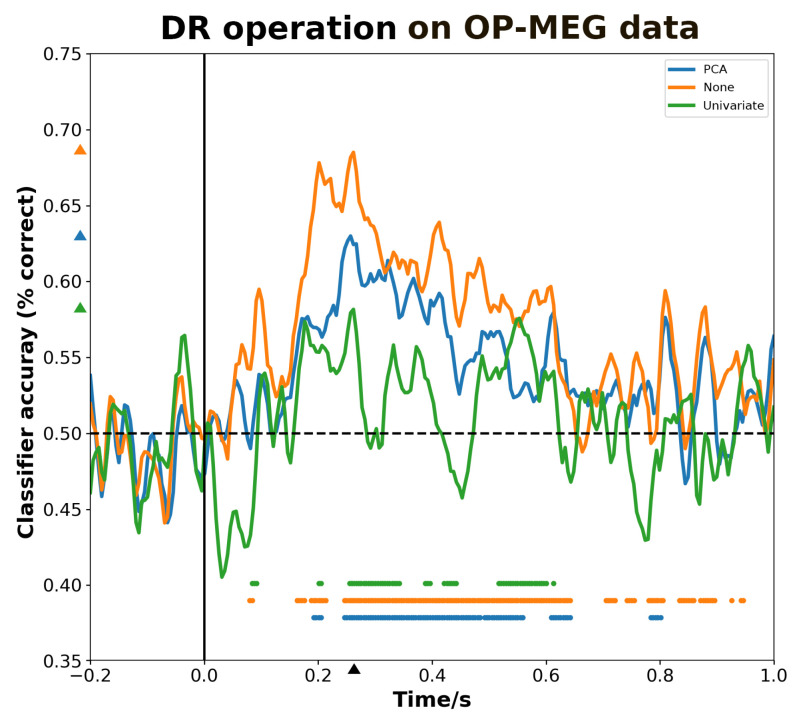
The effect of DR methods on decoding performance. Overall, DR through univariate statistical methods for channel selection of interest is less effective compared to using PCA or no DR operation. Using PCA (blue line) yields the modest gain in performance. The decoding results obtained without any DR operation (orange line) achieved the highest decoding accuracy. The colored triangles marked at *y*-axis denotes the maximum value to the corresponding curves and the triangle at *x*-axis denotes the maximum time. Discs above the *x*-axis indicate the time points where decoding performance is significantly higher than chance.

**Figure 4 bioengineering-11-00609-f004:**
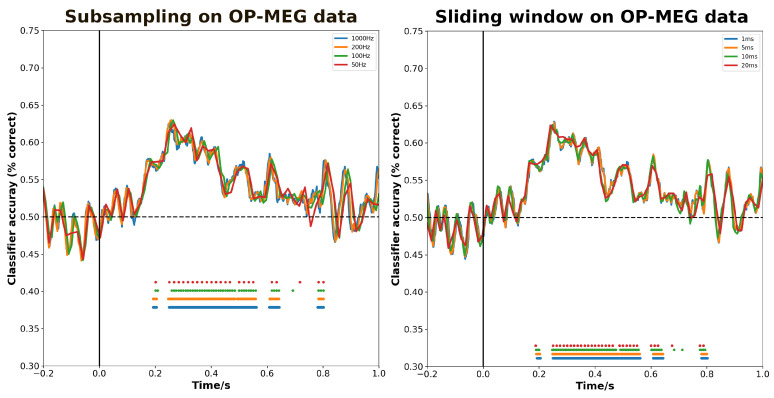
The effect of (**left**) subsampling and (**right**) sliding window approaches to improving SNR on decoding accuracy. The two approaches aimed at improving SNR do not yield a significant enhancement of the decoding performance. Discs above the *x*-axis indicate the time points where the decoding performance is significantly higher than chance.

**Figure 5 bioengineering-11-00609-f005:**
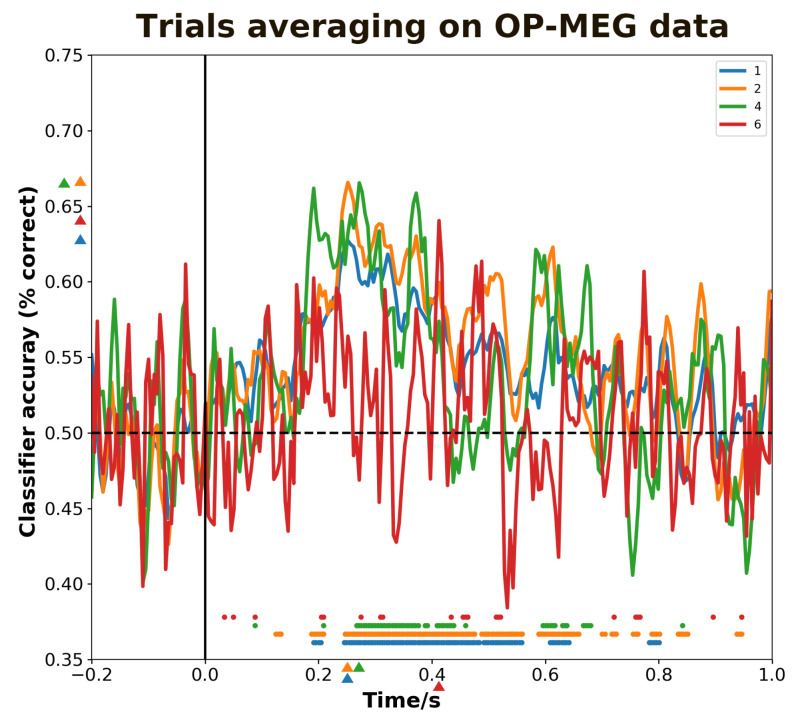
The effect of averaging trials on decoding performance. The blue, orange, green and red curves represent, respectively, the decoding accuracy under four conditions: not averaging and averaging over every 2, 4, 6 epochs. The colored triangles marked at the *y*-axis denote the maximum value to the corresponding curves and the triangles at the *x*-axis denote the corresponding maximum times. Discs above the *x*-axis indicate the time points where decoding performance is significantly higher than chance.

**Figure 6 bioengineering-11-00609-f006:**
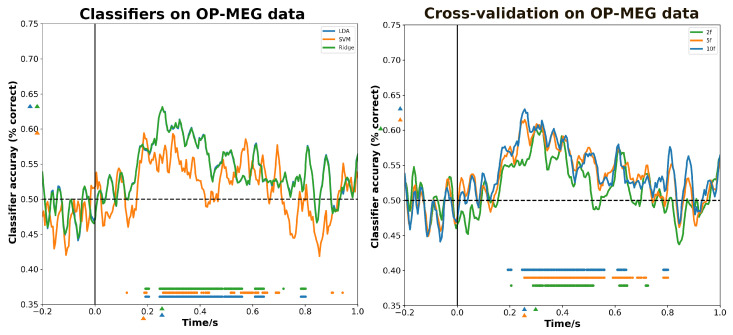
The effect of (**left**) classifier and (**right**) cross-validation approaches on decoding accuracy. The colored triangles marked at the *y*-axis denote the maximum value to the corresponding curves and the triangles at the *x*-axis denote the corresponding maximum times. Discs above the *x*-axis indicate the time points where decoding performance is significantly higher than chance.

**Figure 7 bioengineering-11-00609-f007:**
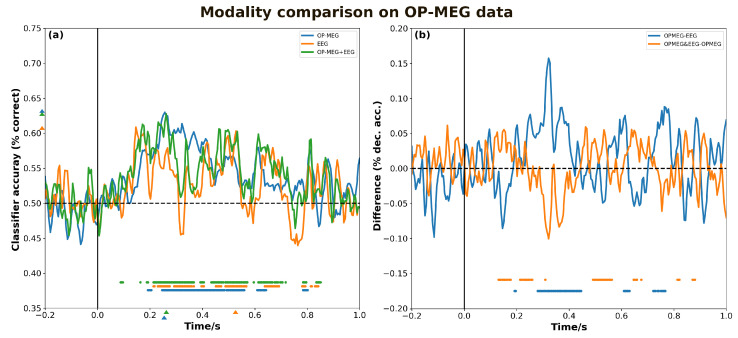
The effect of different modalities on decoding accuracy. (**a**) Grand-averaged time course of decoding for OP-MEG, EEG and OP-MEG and EEG samplings. The colored triangles marked at the *y*-axis denote the maximum value to the corresponding curves and the triangles at the *x*-axis denote the corresponding maximum times. (**b**) Difference curves for the results are shown in (**a**). Discs above the *x*-axis indicate the time points where decoding performance is significantly higher than chance (**a**) and zero (**b**).

**Figure 8 bioengineering-11-00609-f008:**
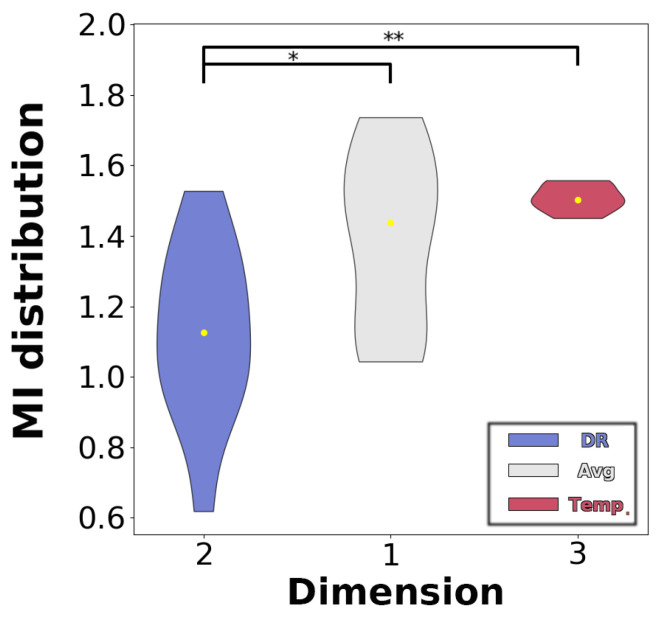
Statistical comparison of MI at different reduction dimensions. The yellow dot in each violin bar indicates the mean value of the data. Notes: * *p* < 0.05; ** *p* < 0.01.

**Figure 9 bioengineering-11-00609-f009:**
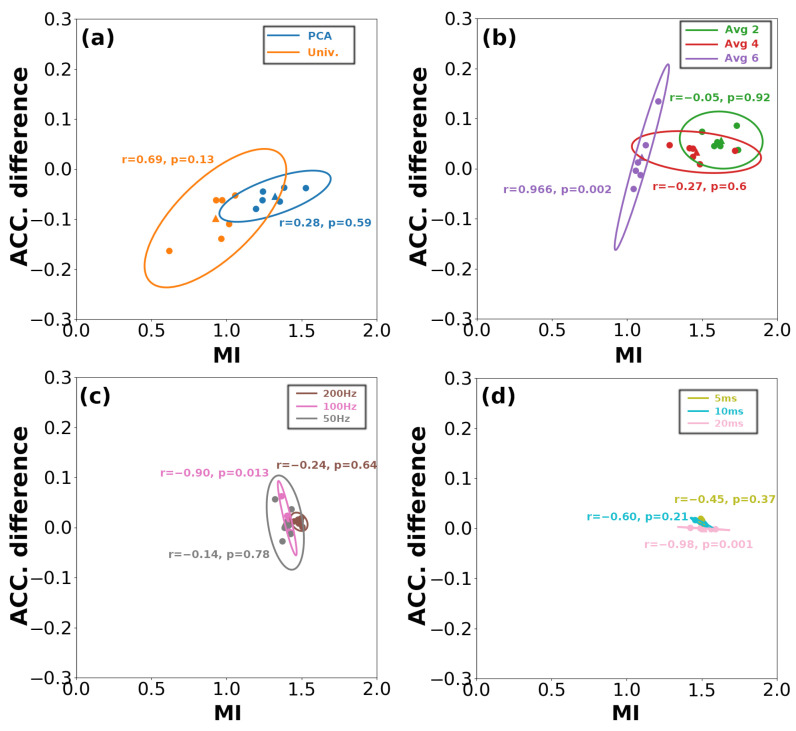
Correlation of the decoding accuracy. (**a**,**b**) represent the correlation results of the MI-computed and corresponding accuracy difference for reduction at the spatial, trial dimension and (**c**,**d**) represent the temporal dimension (resampling and sliding window). The ellipses depicted here represent the covariance confidence, and the triangles represent the mean data value at the *x*- and *y*-axis. ‘r’ and ‘p’ denote the correlation and corresponding significance results. Notes: Univ., univariate; Resamp., resampling; Slid., Sliding window; Temp., temporal; ACC., accuracy.

## Data Availability

The data, aside from the data published in this manuscript, are not publicly available due to privacy restrictions.

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
