# Peer review of "Decoding N400m Evoked Component: A Tutorial on Multivariate Pattern Analysis for OP-MEG Data"

_bioengineering, 2024, doi:10.3390/bioengineering11060609_

Round 1

Reviewer 1 Report

Comments and Suggestions for Authors

The authors discussed neuroimaging studies focusing on pattern analysis and decoding models using fMRI, EEG, and MEG techniques. The paper is well-organized and written. The proposed methodology is significant, and the disclosed results are useful. I think that the paper is acceptable. However, to further improve the already quality of the work, I would suggest incorporating the given improvements

1.     The article mentions the importance of data preprocessing steps such as data transformation and dimensionality reduction. Improving the clarity and detail of these steps could enhance the overall quality of the research.

2.     The decoding and testing processes, including the use of pattern classifiers and cross-validation, could be further elaborated for better understanding and reproducibility.

3.     The impact of varying parameters on decoding results was explored, but the interactions between parameters and their effects on outcomes should be more thoroughly discussed for a clearer interpretation of the results.

4.     The introduction section must end with the structure of the paper.

5.     The article has no literature Review Section. Please add a Literature review section in the paper to prove that before and during the writing of the paper, the authors have studied the concerned and the latest research.

6.     I think the authors used some referencing tools for referencing. Various references are incomplete. Please revisit all the references to add missing details.

7.     The references in this manuscript are not sufficient. In order to catch on the state-of-the-art research situation, please cite more related references and publications published in the past three years.

8.     Please remove plagiarism from the article

Reviewer 2 Report

Comments and Suggestions for Authors

The manuscript proposes an approach to decoding N400m evoked component. In terms of topic, it is well suited to the journal. In terms of methodological approach and presentation, the manuscript suffers from certain drawbacks. I propose to the authors to consider the possibility to address the following remarks.

Remarks:

1. The goal of the reported study is not clearly explained. For example, in the second paragraph of Introduction, the authors state that “the goal of this work is to provide a tutorial guide for the OP-MEG time series data for cognitive neuroscience research” (l. 45-46) and they briefly mention the difference between typical research tasks in BCI and neuroscience research. However, two points are not clearly explained. First, it is not clear to which particular aspect of part of OP-MEG time series data the manuscript relates. And second, is it not clear to which particular aspect of  neuroscience research the manuscript relates. In addition, a similar observation holds for the last paragraph of Introduction.

2. The event-related potential N400 has not been explained in the manuscript. However, to enable the general reader to understand the experimental settings, the authors should have explained this notion. For example, one of the interpretations of N400 is the following: At the functional level, the N400 amplitude reflects the retrieval of lexical information from long-term memory to working memory. At the signal level, the N400 effect is a negative deflection of the signal peaking at about 400 ms after the onset of a stimulus. Without such an explanation, the general reader is not able to understand how the subjects can evaluate the correctness of the final verbs in the stimuli (i.e., selection between the left and right keys), and why one-second pause has been made, etc.

3. The manuscript does not clearly state the scientific contribution.

4. The authors state that “Epochs containing large muscle artifacts were regarded as bad by computing z-score value, and the threshold were set depending on the score distribution” (145-147). How the threshold was determined?

5. In Fig. 3-7, the authors consider classification accuracy. However, they did not explain which classifier was applied.

6. The authors consider three machine learning models: linear discriminant analysis, ridge regression and support vector machine. None of these models is specifically designed for time-series data. The authors should explain why did they select these models as appropriate for this particular task..

7. The following statement is not clear: “The combined decoding to the fused data requires first balancing the number of trials for the respective subjects under the two conditions, and then standardizing the different modality data to scale to unit variance” (246-249).

8. The abbreviations “DR” method and “MI” measure are not introduced.

9. The manuscript would benefit from proofreading (e.g., “From the studies above, Conducting decoding”, l.42-43, etc.).

Comments on the Quality of English Language

Please cf. Remark 9 in the review report.

Round 2

Reviewer 1 Report

Comments and Suggestions for Authors

The authors have successfully incorporated the comments of the reviewer.

No more changes are required in the article.

One thing that is required to be reexamined is the similarity index. Before final submission, please reduce more similarity index in the article.

Reviewer 2 Report

Comments and Suggestions for Authors

The authors have addressed most of the remarks from my previous review report and I believe that the manuscript has been sufficiently improved to warrant publication.